# Virtual Screening, Synthesis, and Biological Evaluation of Some Carbohydrazide Derivatives as Potential DPP-IV Inhibitors

**DOI:** 10.3390/molecules28010149

**Published:** 2022-12-24

**Authors:** Prerana B. Jadhav, Shailaja B. Jadhav, Mehrukh Zehravi, Mohammad S. Mubarak, Fahadul Islam, Philippe Jeandet, Sharuk L. Khan, Nazmul Hossain, Salma Rashid, Long Chiau Ming, Md. Moklesur Rahman Sarker, Mohd Fahami Nur Azlina

**Affiliations:** 1S.N.D. College of Pharmacy, Babhulgaon, Yeola 423401, India; 2PES’s Modern College of Pharmacy, Nigdi, Pune 411044, India; 3Department of Clinical Pharmacy Girls Section, Prince Sattam Bin Abdul Aziz University Alkharj, Al-Kharj 11942, Saudi Arabia; 4Department of Chemistry, The University of Jordan, Amman 11942, Jordan; 5Department of Pharmacy, Faculty of Allied Health Sciences, Daffodil International University, Dhaka 1207, Bangladesh; 6Research Unit-Induced Resistance and Plant Bioprotection, University of Reims, EA 4707—USC INRAe 1488, SFR Condorcet FR CNRS 3417, 51687 Reims, France; 7Department of Pharmaceutical Chemistry, N.B.S. Institute of Pharmacy, Ausa 413520, India; 8School of Medical and Life Sciences, Sunway University, Bandar Sunway 47500, Malaysia; 9Department of Pharmacy, State University of Bangladesh, 77 Satmasjid Road, Dhanmondi, Dhaka 1205, Bangladesh; 10Health Med Science Research Network, 3/1, Block F, Lalmatia, Dhaka 1207, Bangladesh; 11Department of Pharmacology, Faculty of Medicine, University Kebangsaan Malaysia, Jalan Yacob Latif, Kuala Lumpur 56000, Malaysia

**Keywords:** DPP-IV, in vivo, carbohydrazide, 2P8S, ADMET, molecular docking

## Abstract

Dipeptidyl peptidase-4 (DPP-IV) inhibitors are known as safe and well-tolerated antidiabetic medicine. Therefore, the aim of the present work was to synthesize some carbohydrazide derivatives (**1a**–**5d**) as DPP-IV inhibitors. In addition, this work involves simulations using molecular docking, ADMET analysis, and Lipinski and Veber’s guidelines. Wet-lab synthesis was used to make derivatives that met all requirements, and then FTIR, NMR, and mass spectrometry were used to confirm the structures and perform biological assays. In this context, in vitro enzymatic and in vivo antidiabetic activity evaluations were carried out. None of the molecules had broken the majority of the drug-likeness rules. Furthermore, these molecules were put through additional screening using molecular docking. In molecular docking experiments (PDB ID: 2P8S), many molecules displayed more potent interactions than native ligands, exhibiting more hydrogen bonds, especially those with chloro- or fluoro substitutions. Our findings indicated that compounds **5b** and **4c** have IC_50_ values of 28.13 and 34.94 µM, respectively, under in vitro enzymatic assays. On the 21st day of administration to animals, compound **5b** exhibited a significant reduction in serum blood glucose level (157.33 ± 5.75 mg/dL) compared with the diabetic control (Sitagliptin), which showed 280.00 ± 13.29 mg/dL. The antihyperglycemic activity showed that the synthesized compounds have good hypoglycemic potential in fasting blood glucose in the type 2 diabetes animal model (T2DM). Taken all together, our findings indicate that the synthesized compounds exhibit excellent hypoglycemic potential and could be used as leads in developing novel antidiabetic agents.

## 1. Introduction

Hyperglycemia and many other alterations of carbohydrate and protein metabolisms are symptoms of diabetes mellitus, often known as “DM”, a chronic metabolic disorder. The two main causes of diabetes are insulin resistance (in type 1 DM) and an inadequate amount of insulin being produced and used by the body (type 2 DM) [1]. The number of people diagnosed with diabetes every year is increasing across the world. According to estimates provided by the International Diabetes Federation, the number of people living with diabetes reached 30 million in 1985, and this number is expected to reach 380 million by 2025 [2].

When managing type 2 diabetes, the family of oral antidiabetic medicines known as dipeptidyl peptidase-4 (DPP-IV) inhibitors is well-established [3]. Since the publication of the first clinical research evaluating the effects of DPP-IV inhibitors in 2002, multiple randomized controlled trials (RCTs) have shown that DPP-IV inhibitors are effective, safe, and well-tolerated [4]. They achieve this without posing an inherent danger of hypoglycemia by raising endogenous glucagon-like peptide 1 (GLP-1) concentrations, which cause stimulation of insulin release while inhibiting glucagon secretion. Their safety profile seems promising, and their potential effectiveness in bringing down HbA1c is 0.5 to 1.0 percentage points [5]. DPP-IV inhibitors do not affect body weight, and there is evidence that they are safe for the cardiovascular system. These drugs may still be utilized even in patients with compromised renal function [6,7]. In individuals who do not need diabetes treatment with proven cardiovascular benefits, guidelines recommend the extra use of DPP-IV inhibitors if metformin fails to control the patient’s blood sugar levels.

Recently, DPP-IV inhibitors are progressively replacing sulfonylureas as second-line therapy after metformin failure. Additionally, various metformin/DPP-IV inhibitor fixed dosage combinations are already available. In the latter stages of type 2 diabetes, DPP-IV inhibitors are also indicated in the recommendations as part of triple treatments with metformin and SGLT-2 inhibitors or with metformin and insulin [8,9]. These triple therapies are used to treat diabetes. When starting treatment with GLP-1 receptor agonists, the use of DPP-IV inhibitors should be ceased. DPP-IV inhibitors have the potential to be employed as monotherapy in situations when metformin is not appropriate or intolerable. In this regard, a few trials have shown that early combined treatment with metformin and DPP-IV inhibitors benefits certain groups of patients [5,10].

Sitagliptin was the first agent to be launched (in 2006), and additional drugs came out shortly after that. The medications sitagliptin, linagliptin, vildagliptin, saxagliptin, and alogliptin are the most used. Countries in Asia make use of anagliptin, gemigliptin, and teneligliptin as diabetes medications. Moreover, DPP-IV inhibitors are included as part of the therapy algorithms in several national and global recommendations for treating type 2 diabetes [11]. In light of the findings described above, new DPP-IV inhibitors are now being developed. When looking at the structures of approved DPP-IV inhibitors, it was observed that ring systems containing nitrogen atoms along with an amide (–CONH) core were very common in most of them. In view of this, pyrimidine was selected as the basal nucleus to develop DPP-IV inhibitors. Figure 1 pictures the general method used to obtain some of these DPP-IV inhibitors along with their structures. Table 1 lists the different substitutions employed to build the derivatives. In the current study, our goal was to obtain potent DPP-IV inhibitors by synthesizing 6-methyl-2-oxo-4-substituted-*N*’-[(*E*)substituted-methylidene]-1,2,3,4-tetrahydropyrimidine-5-carbohydrazide derivatives. This includes simulations using molecular docking, ADMET analysis, and Lipinski and Veber’s guidelines for the hypothesized derivatives. The derivatives that effectively met all requirements were subjected to wet-lab synthesis and biological testing. Finally, in vitro enzymatic tests and in vivo antidiabetic activity evaluations were carried out.

## 2. Results and Discussion

### 2.1. Virtual Screening of the Designed Derivatives

#### 2.1.1. In Silico ADMET Analysis

Pharmacokinetic characteristics are essential for drug development because they enable researchers to assess the biological components of effective medicines. Lipinski’s rule of five and Veber’s criterion was used to determine if the compounds were appropriate for oral bioavailability (Table 2). All proposed compounds were studied for their ADME features to get a deeper comprehension of their pharmacokinetic profiles and the drug-like qualities each of them (Table 3). Oral acute toxicity, LD_50_ (in mg/kg), toxicity class, and other toxicity parameters have all been anticipated (Table 4).

In developing new medications, the goal is to convert a therapeutic chemical into a dosage form suitable for administering to patients. The pharmacological effects of a medicine, which must occur at the site of action and then disappear after a specific amount of time, are preferable to the once-daily administration of a drug. By analyzing a new drug’s absorption, distribution, metabolism, and excretion (ADME) characteristics, one can make risk-based assessments of the medication’s potential for adverse effects [12,13,14]. This contributes to understanding the course of the pharmacokinetic processes and helps in their investigation. We have synthesized a few 6-methyl-2-oxo-4-substituted-*N’*-[(*E*)substituted-methylidene]-1,2,3,4-tetrahydropyrimidine-5-carbohydrazide derivatives by using a rational drug design technique, that can block the activity of DPP-IV. The ADMET study was carried out on each of the newly generated compounds.

All the molecules’ log P values fell within the range of 1.61 to 3.29, which suggests that an optimal amount of lipophilicity was achieved. The Log P value evaluates the permeability of a drug in the body enabling it to reach the target tissues [15]. All of the investigated compounds had molecular weights of less than 500 Da, suggesting that they can easily move across biological membranes [16,17]. To our relief, none of the compounds negatively impacted the validity of the Lipinski Rule of Five. It has been discovered that the total polar surface area (TPSA) and the number of rotatable bonds can differentiate between compounds that are active and that are not when taken orally. Veber’s rule dictates that the TPSA should be less than 140 Å^2^, and the number of rotatable bonds should be less than 10. None of the synthesized molecules broke Veber’s law in any way. It is an indication that these compounds can be converted into dosage forms that can be taken orally. Despite the fact that many drugs available on the market violate Rule 5, they are still approved by the FDA for oral use. Care must therefore be taken in screening molecules according to the Rule of Five, and a detailed report can be found at https://www.nature.com/articles/nrd.2018.197 (accessed on 16 November 2022).

The pharmacokinetics and drug-likeness features of each molecule were analyzed and estimated so that additional improvements might be made to these compounds. There was, for instance, no evidence that any of these chemicals could cross the blood-brain barrier (BBB). The values of log *Kp* (skin penetration, cm/s) and bioavailability of all the compounds were well within the scope of acceptable ranges. It was observed that all investigated compounds exhibit a high level of gastrointestinal (GI) absorption. Except for the native ligand, none of the compounds broke any of the norms set out by the Ghose, Egan, and Muegge rule. Numerous compounds have been discovered to be inhibitors of cytochrome enzymes, which suggests that these chemicals will likely interfere with the metabolism of other medications. Even more fascinating is that each of the synthesized compounds was a P-gp substrate inhibitor, which suggests that they can prevent drug efflux from the cell. Moreover, acute toxicity predictions showed that all of these compounds belong to class IV of toxicity, which indicates that they are harmful if swallowed (300 < LD50 ≤ 2000) [18].

#### 2.1.2. Molecular Docking

From the very first screening to the application of the Lipinski and Veber rules, the ADMET calculations, and the drug-likeness attributes, none of the molecules had broken the majority of those rules. As a result, these molecules underwent additional screening through molecular docking. Many compounds have demonstrated more powerful interactions and higher binding energies with the target than the native ligand. Table 5 presents details of the interactions and docking scores of the most potent molecules with the DPP-IV enzyme. Furthermore, Table 6 describes the 2D and 3D docking postures of the most potent compounds (docking interactions, docking scores, and binding poses of resting molecules are depicted in the Appendix A).

Molecular docking on the human DPP-IV enzyme was used to get further insight into the structure-activity connection of our new derivatives through the molecules’ binding mode. The lowest energy conformers of all the ligands were docked onto the active site grid of the enzyme. The native ligand had a binding free energy of −9.1 kcal/mol and established just one conventional hydrogen bond with TYR662 through the N–H of the amide group. It formed electrostatic bonds with ARG125 and ARG358, and hydrophobic bonds with ARG358 and PHE357.

All investigated compounds demonstrated good binding affinity ranging from −8.9 to −8.0 kcal/mol. From series **1a**–**1d**, molecule **1b** showed the most potent interactions with a docking score of −9.4 kcal/mol. It formed five conventional hydrogen bonds and one carbon-hydrogen bond with SER349, SER376, THR351, and GLU378. It additionally displayed two electrostatic (attractive charge) bonds with GLU378. It formed two Pi-donor hydrogen bonds with THR350 and SER376 and exhibited hydrophobic interactions with PHE396, CYS394, and VAL354. From the series **2a**–**2d**, molecule **2b** exhibited the most potent binding affinity with a docking score of −9.3 kcal/mol and formed two conventional hydrogen bonds with ARG560 and ASN562. It also formed numerous hydrophobic interactions (Pi-sigma, Pi-Pi T-shaped, alkyl, and Pi-alkyl) with LYS512, ILE529, ARG560, PHE559, PRO475, ALA564, and PHE559.

In the series **3a**–**3d**, compound **3b** exhibited potent interactions and showed −9.3 kcal/mol binding affinity. It formed only one conventional hydrogen bond with HIS740 and two electrostatic (attractive charge) bonds with GLU205. It displayed numerous hydrophobic interactions (Pi-Pi stacked, Pi-Pi T-shaped, amide Pi-stacked, alkyl, and Pi-alkyl) with PHE357, TYR662, TYR547, TYR666, SER630; TYR631, LYS554, VAL656, and TYR547. In the series **4a**–**4d**, compound **4a** showed the highest binding affinity with the target. It exhibited –9.5 kcal/mol binding affinity and formed only two carbon-hydrogen (halogen, F) bonds with PHE559 and VAL558. It formed hydrophobic interactions with LEU514, PRO510, LYS512, and ILE529. On the other hand, compound **5b** from the series **5a**–**5d** exhibited the highest interactions and displayed –9.6 kcal/mol binding affinity. It formed three conventional hydrogen bonds with ILE102, HIS100, and ILE102. It also formed one fluorine halogen bond with ASN74 and hydrophobic interactions with LYS71, TYR105, ILE76, and ILE102. All the aromatic substitutions displayed hydrophobic interactions with the enzyme cavity. The NH of the imidazole ring system has an important function in binding the derivatives to the enzyme cavity. Although most synthesized molecules displayed more potent interactions than native ligands, many displayed more hydrogen bonds, especially molecules with chloro- or fluoro-substitutions. All the designed derivatives exhibited potent interactions and optimized binding affinity with the target. Therefore, all the designed derivatives were subjected to wet lab synthesis and biological evaluation.

The synthesized derivatives were subjected to spectral analysis by FTIR, NMR, and mass spectrometry to confirm the structures. In step I, ethyl 6-methyl-2-oxo-4-substituted-1,2,3,4-tetrahydropyrimidine-5-carboxylate was synthesized via the Biginelli reaction, which was then reacted with hydrazine hydrate to get 6-methyl-2-oxo-4-substituted-1,2,3,4 tetrahydropyrimidine-5-carbohydrazide, in step II. In step III, 6-methyl-2-oxo-4-substituted-1,2,3,4 tetrahydropyrimidine-5-carbohydrazide was refluxed with different aromatic aldehydes (benzaldehyde, 2-hydroxy benzaldehyde, 2-methoxy benzaldehyde, 4-methoxy benzaldehyde) to yield 6-methyl-2-oxo-4-substituted-*N*’-[(*E*)substituted-methylidene]-1,2,3,4-tetrahydropyrimidine-5-carbohydrazide derivatives (**1a**–**5d**). The purity of the synthesized compounds as well as the completion of the reaction were checked with pre-prepared thin-layer chromatography (TLC) plates using *n*-hexane: ethyl acetate (7:3, 8:2, 6:4) as a mobile phase, and iodine vapor as a visualizing agent.

### 2.2. In vitro DPP-IV Enzyme Assay

All the synthesized compounds were subjected to in vitro enzyme assays and the results are shown in Table 7.

Results revealed that all of these compounds exhibit excellent in vitro inhibitory potential. Of all the synthesized molecules, **4c** (which possesses a 2-fluoro phenyl substitution as –Ar position and a -2-methoxy phenyl substitution as –Ar’) showed 60.40 ± 0.08% inhibition at 50 µM and a 34.94 µM IC_50_ value, while compound **5b** (which possesses a -2,4-difluoro phenyl substitution as –Ar position whereas -2-hydroxy phenyl substitution as –Ar’) displayed 70.70 ± 0.37% inhibition at 50 µM and an IC_50_ value of 28.13 µM. Compound **4c** has a fluoro group that withdraws electrons; this makes it more likely to bind to the active site of the enzyme DPP-IV. Due to the presence of an electron-donating methoxy group at Ar’ of compound **4c**, electron density increases on the pyrimidine 2-one ring, which increases the lipophilicity and activity of the compound. Similarly, compound **5b** possesses an electron-withdrawing fluoro group and an electron-donating hydroxyl group. The NH of the carbohydrazones **4c** and **5b** forms hydrogen bonds with the amino acids of the enzyme DPP-IV’s active site. We incorporated various steric, electronic, and hydrophobic groups in the basic nucleus. The synthesized compounds thus showed promising in vitro DPP-IV inhibitory activity.

### 2.3. In Vivo Antidiabetic Activity

Compounds **1b**, **4c**, and **5b** were chosen for further in vivo antidiabetic research based on virtual screening, in vitro enzymatic assay results, and ultimately to better understand structure-activity relationships. Results obtained from in vivo evaluations of the antihyperglycemic activity of the synthesized compounds are displayed in Table 8. These results show the effect of the selected compounds on fasting blood glucose levels compared to sitagliptin. The graph of the in vivo antidiabetic activity is given in Figure 2.

The DPP-IV inhibitor sitagliptin is a fast-acting, strongly bound, reversible, and competitive inhibitor. Long-term inhibition of >70% after 8 h was demonstrated in ob/ob mice at 10 mg/kg (p.o.). In the same model, duration was correlated with long-lasting effectiveness (35% glucose excursion at 8 h) [19]. Therefore, in the present investigation, a 10 mg/kg dose of sitagliptin was employed. The compounds showed significant hypoglycemic potential in fasting blood glucose in an animal model. After the trial, blood glucose levels were considerably higher in diabetes-control animals than in normal control ones. This difference was dramatically reduced in the diabetic treatment groups (compound **5b** at 50 mg/kg and standard treatment with sitagliptin at 10 mg/kg). The serum blood glucose level recovered with compound **5b** was significantly lower on day 21 (157.33 ± 5.75 mg/dL) than that of the diabetic control group (280.00 ± 13.29 mg/dL), whereas that of regular sitagliptin was higher (133.50 ± 11.80 mg/dL). As far as in vitro enzyme assay results are concerned, compound **5b** exhibited an IC_50_ value that is >1400 times higher than that of sitagliptin, though, in animals, **5b** displayed a very potent activity that was significantly comparable with the sitagliptin. This may be linked to the fact that **5b** can inhibit other rate-limiting enzymes contributing to diabetes. This point can be further investigated by performing computational analysis and other in vitro enzyme assays. We are thus aiming to develop more anti-diabetic derivatives considering **5b** as a leading nucleus.

The SAR analysis of all the derivatives revealed important insights into essential structural requirements for effective DPP-IV inhibition. Phenyl ring substitutions are necessary for binding at two sites of the DPP IV enzyme. Removal of the 2-hydroxy substitution of the phenyl ring decreases the inhibitory activity of the derivatives. Hence, the introduction of a polar substituent is essential for the activity. In addition, a significant activity increase was observed due to the structural features of the synthesized compounds. The different structural characteristics, such as the electron-withdrawing groups –Cl and –F, make it easier for the synthesized compounds to connect to the active sites of DPP-IV. In contrast, electron-donating groups such as –OH, –CH_3_, and –OCH_3_ increase electron density on the derivatives and increase lipophilicity, thus enhancing the activity of the synthesized compounds.

## 3. Materials and Methods

### 3.1. Virtual Screening of the Designed Derivatives

#### 3.1.1. In Silico ADMET Analysis

The Lipinski rule of five and the pharmacokinetic properties of the designed molecules were calculated with the help of the molinspiration and SwissADME servers [20,21]. ProTox-II is a web server that is freely accessible to the public and used to carry out an in silico toxicity prediction of the proposed derivatives (http://tox.charite.de/protox_II accessed on 20 October 2022) [18,22,23].

#### 3.1.2. Molecular Docking

The molecular docking was performed using Autodock Vina 1.1.2 in PyRx 0.8 [24]. The process of ligands and protein purification is described in our previously published paper [25]. The Universal Force Field (UFF) has been assigned to carry out energy reduction and optimization [26]. Figure 3 depicts the structure of the DPP-IV enzyme (PDB ID: 2P8S) acquired from the PDB site. The enzyme structures were refined, purified, and prepared for molecular docking using Discovery Studio Visualizer 2019 [27]. The whole molecular docking approach, including identifying cavities and active amino acid residues, was carried out as reported by Khan et al. [15,28,29,30,31,32].

### 3.2. Chemistry

All chemicals and reagents used throughout this work were purchased from Lab Trading Laboratories in Aurangabad, Maharashtra, India, and employed as received. Thin Layer Chromatography (TLC) was used to monitor the reaction, and spots were observed under UV light and exposed to iodine vapors. Melting points (uncorrected) were determined manually using the open capillary technique. ^1^H and ^13^C NMR spectra were obtained with a 500 MHz spectrometer (JEOL) with CDCl_3_ as a solvent and tetramethyl silane (TMS) as an internal standard. Chemical shifts are expressed in δ units or parts per million (ppm), together with the coupling frequencies as singlet (*s*), double (*d*), triplet (*t*), and multiplet (*m*); ^1^H–^1^H coupling constants are given in Hertz. Mass (*m*/*z*) spectra of the synthesized compounds were acquired with a Shimadzu LC-MS system. IR spectra of the synthesized compounds were recorded with an Agilent Technologies’ Microlab IR Spectrophotometer.

#### 3.2.1. Synthesis of ethyl 6-Methyl-2-oxo-4- -1,2,3,4-tetrahydropyrimidine-5-carboxylate

A mixture of 1 mmol of an aromatic aldehyde, ethyl acetoacetate, and urea, and catalytic amounts of CuCl_2_ were ground together for 5–10 min. The solid mass was left standing overnight and later washed with cold water. The product was dried and recrystallized from ethyl alcohol to obtain the pure product [15,29,33].

#### 3.2.2. Synthesis of 6-Methyl-2-oxo-4-substituted-1,2,3,4 Tetrahydropyrimidine-5-carbohydrazide

The product obtained from step I (ethyl 6-methyl-2-oxo-4-substituted-1,2,3,4-tetrahydropyrimidine-5-carboxylate) (10 mmol) was reacted for 3 h under reflux with 10 mmol of hydrazine hydrate in absolute ethanol. The mixture was then cooled to room temperature. The precipitate was filtered and recrystallized from ethanol to give the expected hydrazide [34].

#### 3.2.3. Synthesis of 6-Methyl-2-oxo-4-substituted-*N*’-((E) Substituted-methylidene)-1,2,3,4-tetrahydropyrimidine-5-carbohydrazide Derivatives

The product obtained from step II (6-methyl-2-oxo-4-substituted-1,2,3,4 tetrahydropyrimidine-5-carbohydrazide) (10 mmol) was reacted for 2 h under reflux with 10 mmol of an aromatic aldehyde and acetic acid in 5 mL of ethanol. The mixture was then allowed to cool. The obtained Schiff bases were collected by filtration and recrystallized from ethanol as solid products. Purification of the synthesized compounds was performed using column chromatography [25,35]. Pre-prepared TLC plates were employed to test the purity of the synthesized compounds and reaction completeness using *n*-hexane: ethyl acetate (7:3) as a mobile phase and iodine vapor as a visualization agent. Depicted in Figure 4 is the suggested reaction pathway for synthesizing all the designed derivatives. The different substitutions with compound codes are listed in Table 1 already. The structures of all the synthesized derivatives are depicted in Figure 5.

Characterization data(*E*)-*N*-Benzylidene-6-methyl-2-oxo-4-phenyl-1,2,3,4-tetrahydropyrimidine-5-carbohydrazide (**1a**).

Molecular formula: C_19_H_18_N_4_O_2_, m.p.: 232–235 °C, Rf value: 0.65. Yield: 62%. MS (*m/z*): 334.37. FT-IR (cm^−1^): 3250.10 (–NH str.); 3080.56 (Ar–CH str.); 1686.35 (–CONH str.); 1587.78 (C=N str.); 1235.78 (–C–N str.) ^1^H-NMR (500 MHz, CDCl_3_): δ 7.27, 7.34, 7.39, 7.45, 7.57, 7.62, 7.75, 7.79, 7.85 (*m*, 10H of C_6_H_5_); 7.86, 7.87 (*d*, 2H OF –NH); 8.69 (*s*, 1H of –NH); 5.12 (*s*, 1H of –CH); 2.40 (*t*, 3H of –CH_3_); 6.37 (*s*, 1H of –CH). ^13^C NMR (500 MHz, CDCl_3_): δ 18.98 (1C of –CH_3_); 149.16 (1C of –CH); 120.45, 121.89, 122.99, 123.67, 124.39, 125.33, 126.99, 127.37, 128.31, 129.58, 130.28, 131.71, 132.84, 133.76, 134.86, 135.61 (12C of –C_6_H_5_); 52.21, 101.11, 150.61, 156.10 (4C of pyrimidine); 164.93 (1C of –C=O).

(*E*)-*N*-(2-Hydroxybenzylidene)-6-methyl-2-oxo-4-phenyl-1,2,3,4-tetrahydropyrimidine-5-carbohydrazide (**1b**).

Molecular formula: C_19_H_18_N_4_O_3_, m.p.: 242–245 °C, Rf value: 0.63. Yield: 55%. MS (*m*/*z*): 350.37. FT-IR (cm^–1^): 3250.56 (–NH str.); 3200–3400 (–OH str.); 3020.35 (Ar–CH str.); 1684.67 (–CONH str.); 1535.45 (C=N str.); 1225.45 (–C–N str.). ^1^H-NMR (500 MHz, CDCl_3_): δ 6.40, 6.48, 6.53, 6.59, 6.63, 6.78, 6.89, 6.93, 7.23, 7.38, 7.48, 7.57, 7.6 (*m*, 9H of C_6_H_5_); 7.72, 7.84 (*d*, 2H of –NH); 8.70 (*s*, 1H of –NH); 11.4 (*s*, 1H of –OH); 5.50 (*s*, 1H of –CH); 2.34 (*t*, 3H of –CH_3_); 6.56 (*s*, 1H of –CH). ^13^C NMR (500 MHz, CDCl_3_): δ 18.32 (1C of –CH_3_); 149 (1C of –CH); 120.19, 121.87, 122.47, 123.89, 124.22, 125.78, 126.88, 127.34, 128.39, 129.31, 130.54, 131.67, 132.89, 133.18, 135.39 (12C of –C_6_H_5_); 52.23, 101.28, 150.42, 156.77 (4C of pyrimidine); 164.28 (1C of –C=O).

(*E*)-*N*-(2-Methoxybenzylidene)-6-methyl-2-oxo-4-phenyl-1,2,3,4-tetrahydropyrimidine-5-carbohydrazide (**1c**).

Molecular formula: C_20_H_20_N_4_O_3_, m.p.: 272–275 °C, Rf value: 0.72. Yield: 54%. MS (*m*/*z*): 364.39. FT-IR (cm^–1^): 3240.56 (–NH str.); 3080.56 (Ar–CH str.); 2830.56 (–CH str.); 1686.75 (–CONH str.); 1557.78 (C=N str.); 1235.78 (–C–N str.); 1030.45 (–C–O–C str.). ^1^H-NMR (500 MHz, CDCl_3_): δ 6.71, 6.82, 6.93, 6.99, 7.18, 7.23, 7.38, 7.39, 7.40, 7.49 (*m*, 9H of C_6_H_5_); 7.71, 7.82 (*d*, 2H of –NH); 8.01 (*s*, 1H of –NH); 6.0 (1H of –CH); 2.11, 2.23, 2.28, 2.31, 2.35, 2.39 (*m*, 6H of –CH_3_). ^13^C NMR (500 MHz, CDCl_3_): δ 18.45 (1C of –CH_3_); 57.25 (1C of –OCH_3_); 149.10 (1C of –CH); 120.09, 121.21, 122.41, 123.78, 124.28, 125.72, 126.31, 127.38, 128.41, 129.24, 130.78, 131.38, 132.48, 133.43, 134.82, 135.28 (12C of –C_6_H_5_); 52.28, 101.73, 150.22, 156.45 (4C of pyrimidine); 164.38 (1C of –C=O).

(*E*)-*N*-(4-Methoxybenzylidene)-6-methyl-2-oxo-4-phenyl-1,2,3,4-tetrahydropyrimidine-5-carbohydrazide (**1d**).

Molecular formula: C_20_H_20_N_4_O_3_, m.p.: 275–277 °C, Rf value: 0.74. Yield: 48%. MS (*m*/*z*): 364.39. FT-IR (cm^−1^): 3245.86 (–NH str.); 3080.56 (Ar–CH str.); 2830.56 (–CH str.); 1686.78 (–CONH str.); 1558.78 (C=N str.); 1235.88 (–C–N str.); 1030.45 (–C–O–C str.). ^1^H-NMR (500 MHz, CDCl_3_): δ 6.78, 6.83, 6.92, 6.99, 7.17, 7.24, 7.31, 7.39, 7.41, 7.46, 7.49 (*m*, 9H of C_6_H_5_); 7.62, 7.85 (*d*, 2H of –NH); 8.11 (*s*, 1H of –NH); 5.34 (*s*, 1H of –CH); 1.20, 1.37, 1.42, 1.57, 1.68, 1.72, 1.89, 199, 2.28, 2.31, 2.40 (*m*, 6H of –CH_3_); 6.50 (*s*, 1H of CH). ^13^C NMR (500 MHz, CDCl_3_): δ 18.60 (1C of –CH_3_); 55.25 (1C of –OCH_3_) 149.45 (1C of –CH); 120.09, 121.28, 122.32, 123.67, 124.37, 125.78, 126.31, 127.47, 128.88, 129.99, 130.29, 131.34, 132.67, 133.47, 134.76, 135.21 (12C C_6_H_5_); 52.19, 101.25, 150.54, 156.28 (4C of pyrimidine); 163.78 (1C of –C=O).

(*E*)-*N*-Benzylidene-4-(2-chlorophenyl)-6-methyl-2-oxo-1,2,3,4-tetrahydropyrimidin-5-carbohydrazide (**2a**).

Molecular formula: C_19_H_17_ClN_4_O_2_, m.p.: 301–304 °C, Rf value: 0.76. Yield: 63%. MS (*m*/*z*): 368.81. FT-IR (cm^–1^): 3240.56 (–NH str.); 3080.65 (Ar–CH str.); 1686.57 (–CONH str.); 1557.78 (C=N str.); 1235.80 (–C–N str.) 1030.10 (C–Cl str.). ^1^H-NMR (500 MHz, CDCl_3_): δ 6.41, 6.56, 6.62, 6.69, 6.72, 6.83, 6.92, 6.99, 7.12, 7.25 (*m*, 9H of –C_6_H_5_); 7.51, 7.85, 8.05 (*t*, 3H of –NH); 5.30 (*s*, 1H of –CH); 2.34 (*t*, 3H of –CH_3_); 6.58 (*s*, 1H of CH). ^13^C NMR (500 MHz, CDCl_3_): δ 18.92 (1C of –CH_3_); 149.21 (1C of –CH); 120.09, 121.19, 122.24, 123.34, 124.13, 125.67, 126.32, 127.46, 128.37, 129.28, 130.44, 132.78, 133.91, 134.19, 135.67 (12C of –C_6_H_5_); 52.28, 101.37, 150.17, 156.45 (4C of pyrimidine); 164.27 (1C of –C=O).

(*E*)-4-(2-Chlorophenyl)-*N*-(2-hydroxybenzylidene)-6-methyl-2-oxo-1,2,3,4-tetrahydropyrimidine-5-carbohydrazide (**2b**).

Molecular formula: C_19_H_17_ClN_4_O_3_, m.p.: 308–310 °C, Rf value: 0.78. Yield: 59%. MS (*m*/*z*): 384.81. FT-IR (cm^−1^): 3290.56 (–NH str.); 3100–3400 (–OH str.); 3020.35 (Ar–CH str.); 1686.75 (–CONH str.); 1535.78 (C=N str.); 1205.45 (–C–N str.); 1025.10 (C–Cl str.). ^1^H-NMR (500 MHz, CDCl_3_): δ 6.22, 6.37, 6.42, 6.55, 6.61, 6.69, 6.74, 6.80, 6.94, 7.21 (*m*, 8H of –C_6_H_5_); 7.41, 7.82, 9.01 (*t*, 3H of –NH); 11.52 (*s*, 1H of –OH); 5.60 (*s*, 1H of –CH); 2.34 (*s*, 3H of –CH_3_); 6.51 (*s*, 1H of CH). ^13^C NMR (500 MHz, CDCl_3_): δ 18.75 (1C of –CH_3_); 149.34 (1C of –CH); 120.18, 121.22, 122.38, 123.45, 124.28, 125.89, 126.27, 127.88, 128.47, 129.31, 130.29, 132.77, 133.67, 134.99, 135.38 (12C of –C_6_H_5_); 52.34, 101.67, 150.77, 156.29 (4C of pyrimidine); 162.38 (1C of –C=O).

(*E*)-4-(2-Chlorophenyl)-*N*-(2-methoxybenzylidene)-6-methyl-2-oxo-1,2,3,4-tetrahydropyrimidine-5-carbohydrazide (**2c**).

Molecular formula: C_20_H_19_ClN_4_O_3_, m.p.: 284–286 °C, Rf value: 0.61. Yield: 54%. MS (*m*/*z*): 398.84. FT-IR (cm^−1^): 3250.16 (–NH str.); 3080.56 (Ar–CH str.); 2830.56 (–CH str.); 1685.95 (–CONH str.); 1557.78 (C=N str.); 1225.68 (–C–N str.); 1030.35 (–C–O–C str.); 1068.89 (C–Cl str.). ^1^H-NMR (500 MHz, CDCl_3_): δ 6.42, 6.53, 6.64, 6.73, 6.84, 6.97, 7.10, 7.23 (*m*, 8H of –C_6_H_5_); 7.62, 7.84, 8.13 (*t*, 3H of –NH); 5.34 (*s*, 1H of –CH); 1.21, 1.34, 1.47, 1.52, 1.64, 1.78, 1.88, 1.92, 2.17, 2.26, 2.36, 2.42, 2.56, 2.64 (*m*, 6H of –CH_3_); 6.52 (*s*, 1H of CH). ^13^C NMR (500 MHz, CDCl_3_): δ 17.80 (1C of –CH_3_); 56.35 (1C of –OCH_3_); 149.45 (1C of –CH); 120.27, 121.78, 122.45, 123.37, 124.87, 125.28, 126.90, 127.38, 128.61, 129.28, 130.28, 131.67, 132.48, 133.56, 134.56, 135.27 (12C of –C_6_H_5_); 52.45, 101.67, 150.28, 156.78 (4C of pyrimidine); 164.28 (1C of –C=O).

(*E*)-4-(2-Chlorophenyl)-*N*-(4-methoxybenzylidene)-6-methyl-2-oxo-1,2,3,4-tetrahydropyrimidine-5-carbohydrazide (**2d**).

Molecular formula: C_20_H_19_ClN_4_O_3_, m.p.: 286–288 °C, Rf value: 0.58. MS (*m*/*z*): 398.84. FT-IR (cm^−1^): 3250.60 (–NH str.); 3080.35 (Ar–CH str.); 2835.65 (–CH str.); 1682.70 (–CONH str.); 1558.58 (C=N str.); 1235.38 (–C–N str.); 1030.45 (–C–O–C str.); 1068.89 (C–Cl str.). ^1^H-NMR (500 MHz, CDCl_3_): δ 6.41, 6.49, 6.58, 6.63, 6.74, 6.82, 6.95, 7.10, 7.18, 7.23, 7.29 (*m*, 8H of –C_6_H_5_); 7.3, 7.4, 8.11 (*t*, 3H of –NH); 6.2 (*s*, 1H of –CH); 1.21, 1.32, 1.39, 1.43, 158, 1.73, 1.84, 1.95, 1.99, 2.18, 2.31, 2.43, 2.51, 2.59 (*m*, 6H of –CH_3_); 6.58 (*s*, 1H of –CH). ^13^C NMR (500 MHz, CDCl_3_): δ 18.90 (1C of –CH_3_); 57.25 (1C of –OCH_3_); 149.45 (1C of –CH); 120.09, 121.28, 122.78, 123.89, 124.45, 125.61, 126.34, 127.82, 128.45, 129.82, 130.89, 131.28, 132.54, 133.82, 134.67, 135.34 (12C of –C_6_H_5_); 52.31, 101.23, 150.71, 156.38 (4C of pyrimidine); 162.61 (1C of –C=O).

(*E*)-*N*-Benzylidene-4-(2,4-dichlorophenyl)-6-methyl-2-oxo-1,2,3,4-tetrahydropyrimidine-5-carbohydrazide (**3a**).

Molecular formula: C_19_H_16_Cl_2_N_4_O_2_, m.p.: 313–315 °C, Rf value: 0.59. Yield: 68%. MS (*m*/*z*): 403.26. FT-IR (cm^−1^): 3240.76 (–NH str.); 3070.86 (Ar–CH str.); 1686.75 (–CONH str.); 1557.80 (C=N str.); 1235.88 (–C–N str.) 1060.90 (C–Cl str.). ^1^H-NMR (500 MHz, CDCl_3_): δ 6.42, 6.51, 6.63, 6.78, 6.85, 6.99, 7.14, 7.27 (*m*, 8H of –C_6_H_5_); 7.93, 8.09, 11.04 (*t*, 3H of –NH); 5.66 (*s*, 1H of –CH); 2.33 (*s*, 3H of –CH_3_); 6.54 (*s*, 1H of –CH). ^13^C NMR (500 MHz, CDCl_3_): δ 18.80 (1C of –CH_3_); 149.45 (1C of –CH); 120.27, 121.89, 122.63, 123.56, 124.67, 125.72, 126.39, 127.48, 128.39, 129.49, 130.21, 131.89, 132.67, 133.27, 134.88, 135.39 (12C of –C_6_H_5_); 52.37, 101.89, 150.26, 156.47 (4C of pyrimidine); 164.28 (1C of –C=O).

(*E*)--4-(2,4-Dichlorophenyl)-*N*-(2-hydroxybenzyldene)-6-methyl-2-oxo-1,2,3,4-tetrahydropyrimidine-5-carbohydrazide (**3b**).

Molecular formula: C_19_H_16_Cl_2_N_4_O_2_, m.p.: 305–307 °C, Rf value: 0.76. Yield: 56%. MS (*m*/*z*): 419.26. FT-IR (cm^−1^): 3240.61 (–NH str.); 3234.56 (–OH str.); 3080.75 (Ar–CH str.); 1685.75 (–CONH str.); 1556.58 (C=N str.); 1225.38 (–C–N str.); 1086.58 (C–Cl str.). ^1^H-NMR (500 MHz, CDCl_3_): δ 6.49, 6.53, 6.69, 6.82, 6.99, 7.10, 7.19, 7.29 (*m*, 7H of –C_6_H_5_); 7.60, 7.84, 8.24 (*t*, 3H of –NH); 11.73 (*s*, 1H of –OH); 5.34 (*s*, 1H of –CH); 2.32 (*s*, 3H of –CH_3_); 6.58 (*s*, 1H of CH). ^13^C NMR (500 MHz, CDCl_3_): δ 18.98 (1C of –CH_3_); 149.38 (1C of –CH); 120.90, 121.78, 122.65, 123.45, 124.78, 125.28, 126.45, 127.99, 128.34, 129.37, 130.29, 131.78, 132.45, 133.29, 134.67, 135.38 (12C of –C_6_H_5_); 52.62, 101.61,150.81, 156.73 (4C of pyrimidine); 161.29 (1C of –C=O).

(*E*)-*N*-(2,4-Diclorobenzylidene)-4-(2-methoxyphenyl)-6-methyl-2-oxo-1,2,3,4-tetrahydropyrimidine-5-carbohydrazide (**3c**).

Molecular formula: C_20_H_18_Cl_2_N_4_O_3_, m.p.: 289–292 °C, Rf value: 0.77. Yield: 57%. MS (*m/z*): 433.28. FT-IR (cm^–1^): 3280.60 (–NH str.); 3080.56 (Ar–CH str.); 2830.56 (–CH str.); 1676.75 (–CONH str.); 1556.68 (C=N str.); 1235.87 (–C–N str.); 1030.45 (–C–O–C str.); 1061.29 (C–Cl str.). ^1^H-NMR (500 MHz, CDCl_3_): δ 6.43, 6.51, 6.64, 6.78, 6.86, 6.92, 6.99, 7.09, 7.19, 7.24 (*m*, 7H of –C_6_H_5_); 7.52, 7.81, 8.11 (*t*, 3H of –NH); 5.38 (1H of –CH); 1.21, 1.37, 1.43, 1.58, 1.68, 1.74, 1.88, 1.98, 2.18, 2.22, 2.39, 2.57, 2.64 (*m*, 6H of –CH_3_); 6.56 (*s*, 1H of –CH). ^13^C NMR (500 MHz, CDCl_3_): δ 18.67 (1C of –CH_3_); 55.25 (1C of –OCH_3_) 149.37 (1C of –CH); 120.09, 121.78, 122.37, 123.84, 124.28, 125.89, 126.27, 127.38, 128.38, 130.48, 132.78, 133.22, 134.56, 135.38 (12C of –C_6_H_5_); 52.34, 101.61, 150.37, 156.29 (4C of pyrimidine); 164.39 (1C of –C=O).

(*E*)-*N*-(2,4-Diclorobenzylidene)-4-(4-methoxyphenyl)-6-methyl-2-oxo-1,2,3,4-tetrahydropyrimidine-5-carbohydrazide (**3d**).

Molecular formula: C_20_H_18_Cl_2_N_4_O_3_, m.p.: 288–292 °C, Rf value: 0.57. Yield: 50%. MS (*m*/*z*): 433.28. FT-IR (cm^−1^): 3260.86 (–NH str.); 3070.56 (Ar–CH str.); 2828.86 (–CH str.); 1687.45 (–CONH str.); 1558.30 (C=N str.); 1225.78 (–C–N str.); 1020.85 (–C–O–C str.); 1068.90 (C–Cl str.). ^1^H-NMR (500 MHz, CDCl_3_): δ 6.41, 6.57, 6.65, 6.71, 6.83, 6.89, 6.99, 7.12, 7.28 (*m*, 7H of –C_6_H_5_); 7.72, 7.94, 8.21 (*t*, 3H of –NH); 5.32 (*s*, 1H of –CH); 1.23, 1.39, 1.48, 1.67, 1.78, 1.98, 2.31, 2.48, 2.59, 2.69 (*m*, 6H of –CH_3_); 6.56 (*s*, 1H of CH). ^13^C NMR (500 MHz, CDCl_3_): δ 18.20 (1C of –CH_3_); 57.35 (1C of –OCH_3_); 149.89 (1C of –CH); 120.09, 121.89, 122.73, 123.28, 125.29, 126.89, 127.39, 128.54, 129.89, 130.29, 131.67, 132.48, 133.90, 134.87, 135.29 (12C of –C_6_H_5_); 52.34, 101.56, 150.13, 156.34 (4C of pyrimidine); 162.28 (1C of –C=O).

(*E*)-*N*-Benzylidene-4-(2-fluorophenyl)-6-methyl-2-oxo-4-phenyl-1,2,3,4-tetrahydropyrimidine-5-carbohydrazide (**4a**).

Molecular formula: C_19_H_17_FN_4_O_2_, m.p.: 282–286 °C, Rf value: 0.59. Yield: 53%. MS (*m*/*z*): 352.36. FT-IR (cm^–1^): 3240.86 (–NH str.); 3050.66 (Ar–CH str.); 1686.22 (–CONH str.); 1557.78 (C=N str.); 1235.87 (–C–N str.); 1102.35 (C–F str.). ^1^H-NMR (500 MHz, CDCl_3_): δ 6.40, 6.48, 6.57, 6.61, 6.78, 6.83, 6.99, 7.18, 7.27, 7.29 (*m*, 9H of –C_6_H_5_); 7.53, 7.86, 8.18 (*t*, 3H of –NH); 5.33 (*s*, 1H of –CH); 2.31 (*s*, 3H of –CH_3_); 6.56 (*s*, 1H of CH). ^13^C NMR (500 MHz, CDCl_3_): δ 17.70 (1C of –CH_3_); 149.45 (1C of –CH); 120.23, 121.41, 122.78, 123.37, 124.72, 125.47, 126.83, 127.34, 128.38, 129.38, 130.45, 131.56, 132.38, 133.67, 134.28, 135.36 (12C of –C_6_H_5_); 52.23, 101.45, 150.37, 156.29 (4C of pyrimidine); 164.38 (1C of –C=O).

(*E*)--4-(2-Fluorophenyl)-*N*-(2-hydroxybenzyldene)-6-methyl-2-oxo-1,2,3,4-tetrahydropyrimidine-5-carbohydrazide (**4b**).

Molecular formula: C_19_H_17_FN_4_O_3_, m.p.: 205–208 °C, Rf value: 0.63. Yield: 58%. MS (*m*/*z*): 368.36. FT-IR (cm^−1^): 3280.45 (–NH str.); 3245.65 (–OH str.); 3020.35 (Ar–CH str.); 1684.67 (–CONH str.); 1535.45 (C=N str.); 1235.45 (–C–N str.); 1062.35 (C–F str.). ^1^H-NMR (500 MHz, CDCl_3_): δ 6.49, 6.59, 6.67, 6.73, 6.86, 6.92, 7.12, 7.19, 7.29 (*m*, 8H of –C_6_H_5_); 7.52, 8.09, 11.05 (*t*, 3H of –NH); 11.83 (1H of –OH); 5.32 (*s*, 1H of –CH); 2.34 (*s*, 3H of –CH_3_); 6.58 (*s*, 1H of CH). ^13^C NMR (500 MHz, CDCl_3_): δ 18.75 (1C of –CH_3_); 149.36 (1C of –CH); 120.56, 121.78, 122.78, 123.67, 124.82, 125.38, 126.87, 126.38, 127.48, 128.99, 129.39, 130.29, 131.56, 132.89, 133.27, 134.78, 135.09 (12C of –C_6_H_5_); 52.23, 101.34, 150.17, 156.81 (4C of pyrimidine); 161.27 (1C of –C=O).

(*E*)--4-(2-Fluorophenyl)-*N*-(2-methoxybenzyldene)-6-methyl-2-oxo-1,2,3,4-tetrahydropyrimidine-5-carbohydrazide (**4c**).

Molecular formula: C_20_H_19_FN_4_O_3_, m.p.: 222–224 °C, Rf value: 0.68. Yield: 62%. MS (*m*/*z*): 382.36. FT-IR (cm^−1^): 3250.68 (–NH str.); 3060.77 (Ar–CH str.); 2820.65 (–CH str.); 1686.80 (–CONH str.); 1557.78 (C=N str.); 1235.78 (–C–N str.); 1025.35 (–C–O–C str.); 962.35 (C–F str.). ^1^H-NMR (500 MHz, CDCl_3_): δ 6.44, 6.53, 6.62, 6.73, 6.89, 6.94, 7.09, 7.19, 7.28 (*m*, 8H of –C_6_H_5_); 7.63, 7.72, 8.09 (*t*, 3H of –NH); 5.34 (*s*, 1H of –CH); 1.20, 1.25, 1.37, 1.45, 1.52, 1.67, 1.78, 1.88, 1.92, 2.09, 2.17, 2.33, 2.49, 2.55, 2.62 (*m*, 6H of –CH_3_); 6.58 (*s*, 1H of CH). ^13^C NMR (500 MHz, CDCl_3_): δ 18.90 (1C of –CH_3_); 55.45 (1C of –OCH_3_); 149.02 (1C of –CH); 120.99, 121.87, 122.67, 123.45, 124.27, 125.89, 126.41, 127.38, 128.81, 129.37, 130.89, 131.28, 132.90, 134.22, 135.78 (12C of –C_6_H_5_); 52.37, 101.28, 150.90, 156.51 (4C of pyrimidine); 164.38 (1C of –C=O).

(*E*)--4-(2-Fluorophenyl)-*N*-(4-methoxybenzyldene)-6-methyl-2-oxo-1,2,3,4-tetrahydropyrimidine-5-carbohydrazide (**4d**).

Molecular formula: C_20_H_19_FN_4_O_3_, m.p.: 313–316 °C, Rf value: 0.67. Yield: 54%. MS (*m*/*z*): 382.36. FT-IR (cm^–1^): 3260.56 (–NH str.); 3080.20 (Ar–CH str.); 2820.16 (–CH str.); 1686.35 (–CONH str.); 1557.80 (C=N str.); 1236.18 (–C–N str.); 1031.40 (–C–O–C str.); 982.35 (C–F str.). ^1^H-NMR (500 MHz, CDCl_3_): δ 6.42, 6.49, 6.52, 6.68, 6.78, 6.82, 6.93, 7.05, 7.15, 7.29 (*m*, 8H of –C_6_H_5_); 7.63, 7.82, 8.25 (*t,* 3H of –NH); 5.34 (*s*, 1H of –CH); 1.22, 1.34, 1.45, 1.51, 1.68, 1.88, 1.98, 2.09, 2.19, 2.28, 2.37, 2.44, 2.58, 2.67 (*m*, 6H of –CH_3_); 6.56 (*s*, 1H of –CH). ^13^C NMR (500 MHz, CDCl_3_): δ 18.25 (1C of –CH_3_); 57.25 (1C of –OCH_3_); 149.34 (1C of –CH); 120.34, 121.89, 122.34, 123.67, 124.98, 125.48, 126.89, 128.49, 129.33, 130.29, 131.87, 133.29, 134.77, 135.64 (12C of –C_6_H_5_); 52.45, 101.28, 150.38, 156.62 (4C of pyrimidine); 163.28 (1C of –C=O).

(*E*)-*N*-Benzylidene-4-(2,4-difluorophenyl)-6-methyl-2-oxo-1,2,3,4-tetrahydropyrimidine-5-carbohydrazide (**5a**).

Molecular formula: C_19_H_16_F_2_N_4_O_2_, m.p.: 303–306 °C, Rf value: 0.49. Yield: 65%. MS (*m*/*z*): 370.35. FT-IR (cm^−1^): 3240.56 (–NH str.); 3080.20 (Ar–CH str.); 1676.85 (–CONH str.); 1556.68 (C=N str.); 1225.35 (–C–N str.) 1062.35 (C–F str.). ^1^H-NMR (500 MHz, CDCl_3_): δ 6.40, 6.48, 6.55, 6.62, 6.79, 6.83, 6.99, 7.03, 7.26, 7.20 (*m*, 8H of –C_6_H_5_); 7.72, 7.83, 8.14 (*t,* 3H of –NH); 5.32 (*s,* 1H of –CH); 2.31 (*s*, 3H of –CH_3_); 6.52 (*s,* 1H of –CH). ^13^C NMR (500 MHz, CDCl_3_): δ 18.67 (1C of –CH_3_); 149.34 (1C of –CH); 120.90, 121.45, 122.67, 123.56, 124.38, 125.89, 126.82, 127.38, 128.49, 129.39, 130.29, 132.89, 133.45, 134.71, 135.45 (12C of –C_6_H_5_); 52.21, 101.39, 150.09, 156.03 (4C of pyrimidine); 164 (1C of –C=O).

(*E*)--4-(2,4-Difluorophenyl)-*N*-(2-hydroxybenzyldene)-6-methyl-2-oxo-1,2,3,4-tetrahydropyrimidine-5-carbohydrazide (**5b**).

Molecular formula: C_19_H_16_F_2_N_4_O_3_, m.p.: 318–321 °C, Rf value: 0.72. Yield: 63%. MS (*m*/*z*): 386.32. FT-IR (cm^−1^): 3290.56 (–NH str.); 3200–3400 (–OH str.); 3005.10 (Ar–CH str.); 1686.75 (–CONH str.); 1557.78 (C=N str.); 1235.78 (–C–N str.); 1105.88 (C–F str.). ^1^H-NMR (500 MHz, CDCl_3_): δ 6.80, 6.89, 6.91, 7.08, 7.14, 7.21, 7.38, 7.45, 7.51, 7.59 (*m*, 7H of –C_6_H_5_); 7.87, 7.52, 8.30 (*t*, 3H of –NH); 11.30 (*s,* 1H of –OH); 5.33 (*s*, 1H of –CH); 2.31 (*s*, 3H of –CH3); 6.58 (*s*, 1H of CH). ^13^C NMR (500 MHz, CDCl_3_): δ 18.75 (1C of –CH3); 149.09 (1C of –CH); 120.89, 121.27, 122.34, 123.67, 124.38, 125.38, 126.78, 127.38, 128.38, 129.89, 130.29, 131.88, 132.56, 133.29, 135.33 (12C of –C_6_H_5_); 52.88, 101.34, 150.41, 156.37 (4C of pyrimidine); 162.45 (1C of –C=O).

(*E*)-*N*-(2,4-Difluorobenzylidene)-4-(2-methoxyphenyl)-6-methyl-2-oxo-1,2,3,4-tetrahydropyrimidine-5-carbohydrazide (**5c**).

Molecular formula: C_20_H_18_F_2_N_4_O_3_, m.p.: 275–278 °C, Rf value: 0.61. MS (*m*/*z*): 400.37. FT-IR (cm^–1^): 3230.86 (–NH str.); 3080.56 (Ar–CH str.); 2825.86 (–CH str.); 1686.65 (–CONH str.); 1557.68 (C=N str.); 1235.68 (–C–N str.); 1030.45 (–C–O–C str.); 1062.35 (C–F str.). ^1^H-NMR (500 MHz, CDCl_3_): δ 6.40, 6.48, 6.55, 6.62, 6.78, 6.82, 6.92, 7.03, 7.19, 7.21 (*m*, 7H of –C_6_H_5_); 7.71, 7.83, 8.30 (*t*, 3H of –NH); 5.32 (1H of –CH); 1.20, 1.31, 1.39, 1.47, 1.51, 1.68, 1.78, 1.89, 1.93, 2.18, 2.27, 2.39, 2.48, 2.55, 2.60 (*m*, 6H of –CH_3_). ^13^C NMR (500 MHz, CDCl_3_): δ 17.98 (1C of –CH_3_); 57.25 (1C of –OCH_3_); 149.34 (1C of –CH); 120.90, 121.28, 122.45, 123.47, 125.38, 126.89, 127.99, 128.39, 129.99, 130.29, 131.77, 132.78, 133.99, 134.28, 135.77 (12C of –C_6_H_5_); 52.34, 101.23, 150.78, 156.44 (4C of pyrimidine); 164.45 (1C of –C=O).

(*E*)-*N*-(2,4-Difluorobenzylidene)-4-(4-methoxyphenyl)-6-methyl-2-oxo-1,2,3,4-tetrahydropyrimidine-5-carbohydrazide (**5d**).

Molecular formula: C_20_H_18_F_2_N_4_O_3_, m.p.: 301–304 °C, Rf value: 0.58. MS (*m*/*z*): 400.37. FT-IR (cm^−1^): 3255.46 (–NH str.); 3050.66 (Ar–CH str.); 2840.65 (–CH str.); 1686.75 (–CONH str.); 1557.78 (C=N str.); 1232.78 (–C–N str.); 1035.50 (–C–O–C str.); 1065.50 (C–F str.). ^1^H-NMR (500 MHz, CDCl_3_): δ 6.41, 6.52, 6.67, 6.78, 6.88, 6.93, 6.98, 7.09, 7.27 (*m*, 7H of –C_6_H_5_); 7.52, 7.71, 8.58 (*t*, 3H of –NH); 5.34 (*s*, 1H of –CH); 1.20, 1.31, 1.42, 1.56, 1.67, 1.76, 1.83, 1.98, 2.18, 2.37, 2.45, 2.58, 2.63 (*m*, 6H of –CH_3_). ^13^C NMR (500 MHz, CDCl_3_): δ 18.80 (1C of –CH_3_); 55.45 (1C of –OCH_3_); 149.78 (1C of –CH); 120.34, 121.89, 122.38, 123.84, 124.77, 124.39, 125.89, 126.29, 127.99, 128.62, 129.77, 130.98, 131.78, 133.87, 134.67, 135.67 (12C of –C_6_H_5_); 52.67, 101.34, 150.23, 156.98 (4C of pyrimidine); 161.45 (1C of –C=O).

### 3.3. In Vitro DPP-IV Enzymatic Assay

The prepared compounds were investigated in vitro for their ability to inhibit the DPP-IV enzyme, and the detailed method is already described in our previously published paper [25]. Sitagliptin was used as a positive control, as a DPP-IV inhibitor. Both the percentage inhibition at 50 µM and IC_50_ values in µM were determined for all the tested compounds [36]. The pipetting summary for performing the DPP-IV enzymatic assay as given in the assay protocol can be accessed at https://cdn.caymanchem.com/cdn/insert/700210.pdf (accessed on 20 October 2022).

### 3.4. In Vivo Antidiabetic Activity

#### 3.4.1. Animals and Ethical Approvals

The antihyperglycemic potential of certain compounds was investigated in Wistar rats. STZ-induced diabetic rats were employed for the antihyperglycemic testing. The animal experiments were approved by the Institutional Animal Ethics Committee (IAEC) in New Delhi, India (Protocol Number: IAEC/MCP/009/2020). During the research period, the Committee for the Purpose of Control and Supervision of Experiments on Animals (CPCSEA) and IAEC criteria for animal care were followed.

#### 3.4.2. Acute Toxicity Studies

Acute toxicity studies were performed as per the OECD guideline 423. For compounds **1b**, **4c**, and **5b**, three groups of six rats weighing 80–120 g were starved overnight and given the test compounds orally at doses of 200, 500, 1000, and 2000 mg/kg body weight. The animals were monitored for 72 h, for symptoms of acute toxicity, such as increased or reduced motor activity, tremors, convulsions, and drowsiness, among others. It was observed that at a dosage of 500 mg/kg body weight, more than half of the animals died. As a result, the OECD-recommended dosage for assessing the antidiabetic action was set at 50 mg/kg (i.e., one-tenth of the 2000 mg/kg body weight) [37].

#### 3.4.3. STZ-Induced T2DM Model

Rats were given a high-fat diet after one week of acclimation. Diabetes was induced in the animals as per the method each described in [25]. Rats were then divided into six groups with six animals. Group I served as normal control with normal rats receiving the vehicle only. Group II served as diabetes control in which rats received the vehicle only. Group III served as standard and received sitagliptin at a dose of 10 mg/kg body weight (b.w.). Groups IV, V, and VI served as test groups and received compounds **1b**, **5b**, and **4c**, respectively, at doses of 50 mg/kg b.w. The diabetic rats were given the tested compound suspended in 0.5% *w*/*v* CMC every day for 21 days. On days 1, 7, 14, and 21, blood samples were taken from the tail veins of each animal that had fasted overnight. A glucometer was used to measure blood glucose levels [38,39].

## 4. Conclusions

In summary, we have designed and developed a few compounds that can inhibit DPP-IV. The 6-methyl-2-oxo-4-substituted-*N*’-((*E*)substituted-methylidene)-1,2,3,4-tetrahydropyrimidine-5-carbohydrazide derivatives (**1a**–**5d**) were synthesized and characterized by spectral analysis. From the very first screening to the application of the Lipinski rule, the Veber rule, the ADMET calculations, and the drug-likeness attributes, none of these molecules had broken the majority of the drug-likeness rules. All the designed derivatives exhibited potent interactions and optimized binding affinity with the target enzyme. Therefore, wet lab synthesis and biological assessment were performed on all the proposed derivatives. In vitro DPP-IV inhibitory effects of the synthesized compounds were promising. The antihyperglycemic activity of the developed compounds in fasting blood glucose in animal models demonstrates that they have excellent hypoglycemic potential. As the developed molecules demonstrated potent in vitro and in vivo activities in the present study, we have concluded that this nucleus can be explored further and treated as a lead nucleus for developing novel antidiabetic agents. SAR analysis of all the derivatives revealed important insights into the essential structural requirements for effective DPP-IV inhibition. Phenyl ring substitutions are necessary for the binding at the two sites of the DPP IV enzyme. Hence, the introduction of a polar substituent is essential for the activity. In addition, a significant increase in activity was observed due to certain structural characteristics of the synthesized compounds. Different structural features, such as electron-withdrawing groups, for example –Cl and –F, facilitate the binding of the synthesized compounds to the active sites of DPP-IV. In contrast, electron-donating groups, such as –OH, –CH_3_, and –OCH_3_, increase the electron density of the derivatives and increase their lipophilicity, thus enhancing the activity of the synthesized compounds. By studying the structural requirements necessary for their activity, researchers can thus develop more carbohydrazide derivatives as potential DPP-IV inhibitors.

## Figures and Tables

**Figure 1 molecules-28-00149-f001:**
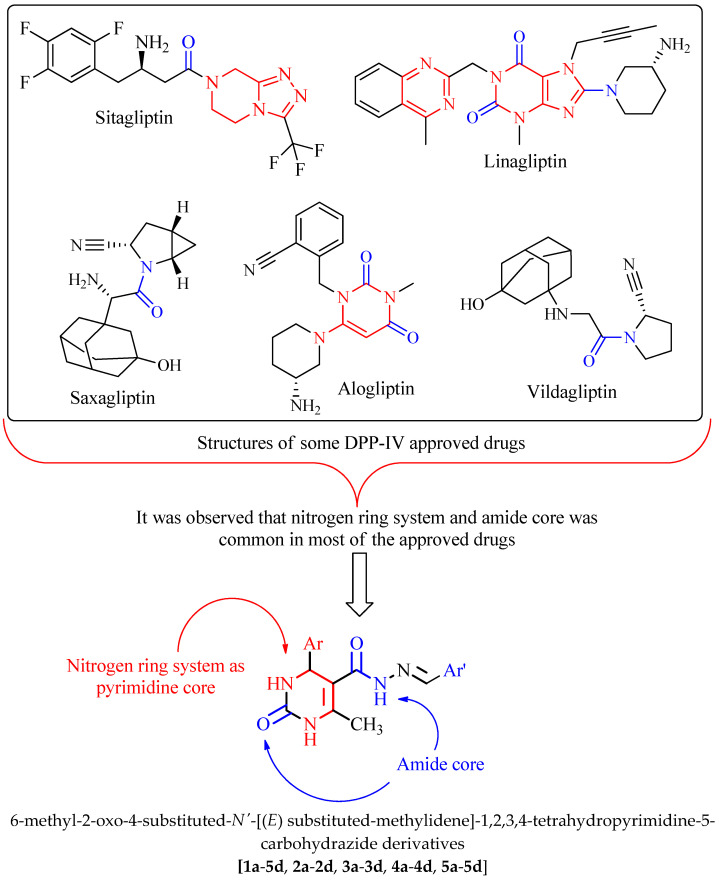
The structures of approved DPP-IV inhibitors and designing approach to design some carbohydrazide derivatives.

**Figure 2 molecules-28-00149-f002:**
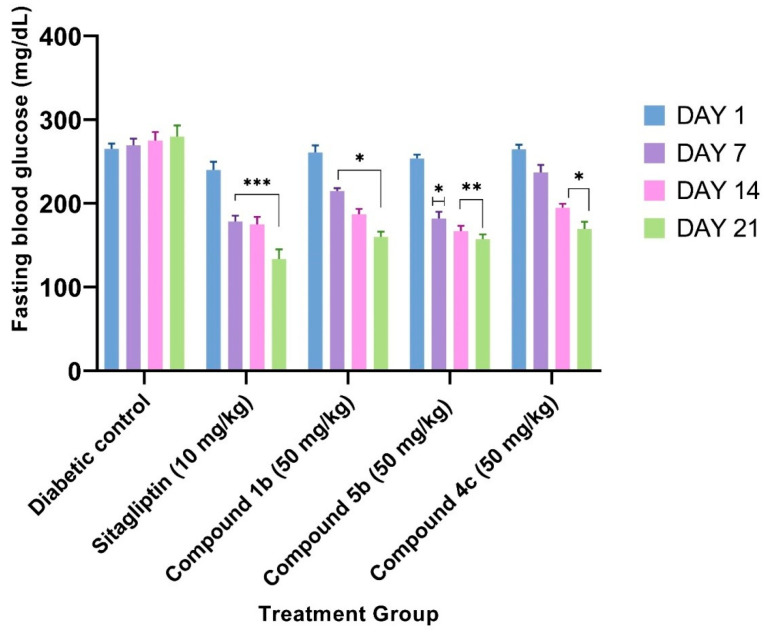
Effect of compounds **1b**, **4c**, and **5b** on blood glucose levels of STZ-induced diabetic rats. The data are expressed as the mean ± S.E.M. (standard error of the mean) of six rats. The levels of significance are indicated as **p* < 0.05; ** *p* < 0.01; and *** *p* < 0.001 when compared to the diabetic control group.

**Figure 3 molecules-28-00149-f003:**
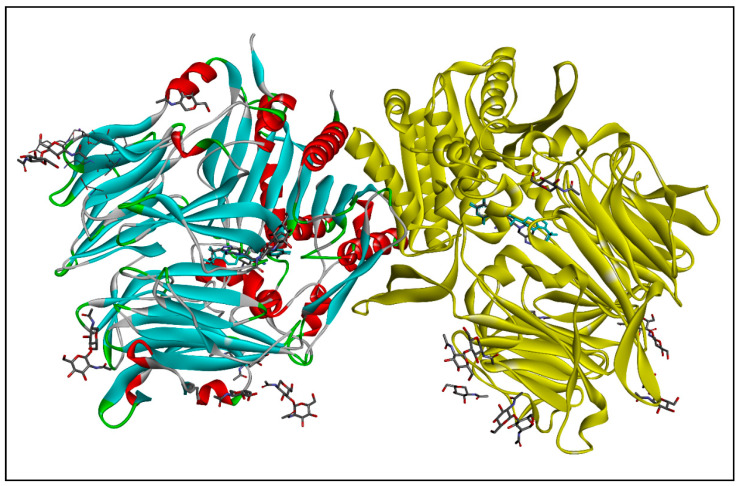
The crystal structure of the human dipeptidyl peptidase IV/CD26 complexed with a cyclohexylamine inhibitor where Chain A: highlighted in multicolor; Chain B: highlighted in yellow color.

**Figure 4 molecules-28-00149-f004:**
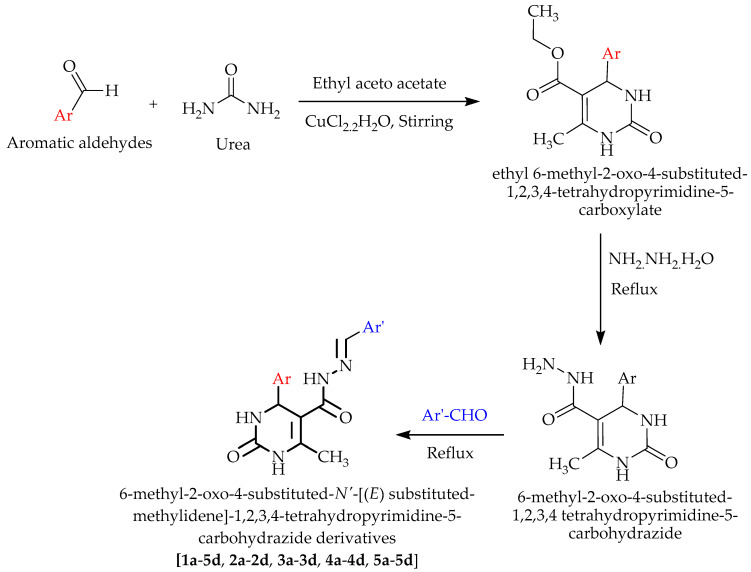
Proposed reaction scheme for the synthesis of carbohydrazide derivatives.

**Figure 5 molecules-28-00149-f005:**
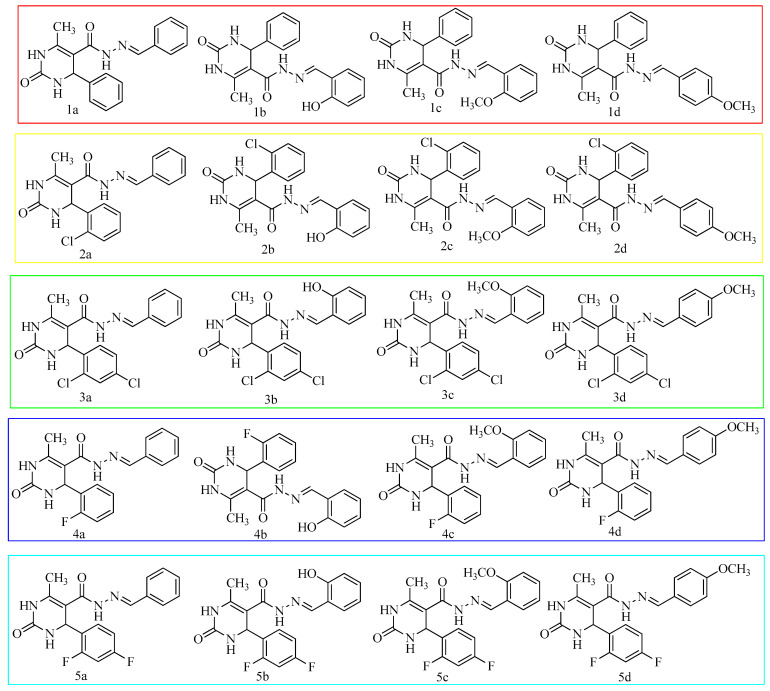
The structures of synthesized derivatives.

**Table 1 molecules-28-00149-t001:** Different substitutions of titled derivatives and compound codes.

Codes	Ar	Ar’	Codes	Ar	Ar’
**1a**	Phenyl	Phenyl	**3c**	2,4-dichlorophenyl	2-methoxyphenyl
**1b**	Phenyl	2-hydroxyphenyl	**3d**	2,4-dichlorophenyl	4-methoxyphenyl
**1c**	Phenyl	2-methoxyphenyl	**4a**	2-fluorophenyl	Phenyl
**1d**	Phenyl	4-methoxyphenyl	**4b**	2-fluorophenyl	2-hydroxyphenyl
**2a**	2-chlorophenyl	Phenyl	**4c**	2-fluorophenyl	2-methoxyphenyl
**2b**	2-chlorophenyl	2-hydroxyphenyl	**4d**	2-fluorophenyl	4-methoxyphenyl
**2c**	2-chlorophenyl	2-methoxyphenyl	**5a**	2,4-difluorophenyl	Phenyl
**2d**	2-chlorophenyl	4-methoxyphenyl	**5b**	2,4-difluorophenyl	2-hydroxyphenyl
**3a**	2,4-dichlorophenyl	Phenyl	**5c**	2,4-difluorophenyl	2-methoxyphenyl
**3b**	2,4-dichlorophenyl	2-hydroxyphenyl	**5d**	2,4-difluorophenyl	4-methoxyphenyl

**Table 2 molecules-28-00149-t002:** Calculations of Lipinski’s rule of five and Veber’s rule for carbohydrazide derivatives (**1a**–**5d**).

Compound Codes	Lipinski Rule of Five					Veber’s Rule	
	Log P (<5)	Mol. Wt. (<500 Da)	HBA (<10)	HBD (<5)s	Violations	Total Polar Surface Area (Å^2^) (<140 Å^2^)	No. of Rotatable Bonds (<10)
NL	3.29	419.37	10	01	0	59.97	03
**1a**	1.97	334.37	3	3	0	82.59	5
**1b**	1.61	350.37	4	4	0	102.82	5
**1c**	2.1	364.4	4	3	0	91.82	6
**1d**	2.07	364.4	4	3	0	91.82	6
**2a**	2.44	368.82	3	3	0	82.59	5
**2b**	2.06	384.82	4	4	0	102.82	5
**2c**	2.61	398.84	4	3	0	91.82	6
**2d**	2.59	398.84	4	3	0	91.82	6
**3a**	2.98	403.26	3	3	0	82.59	5
**3b**	2.58	419.26	4	4	0	102.82	5
**3c**	3.16	433.29	4	3	0	91.82	6
**3d**	3.12	433.29	4	3	0	91.82	6
**4a**	2.24	352.36	4	3	0	82.59	5
**4b**	1.86	368.36	5	4	0	102.82	5
**4c**	2.41	382.39	5	3	0	91.82	6
**4d**	2.38	382.39	5	3	0	91.82	6
**5a**	2.56	370.35	5	3	0	82.59	5
**5b**	2.19	386.35	6	4	0	102.82	5
**5c**	2.73	400.38	6	3	0	91.82	6
**5d**	2.7	400.38	6	3	0	91.82	6

Where: NL, Native ligand; Mol. Wt., molecular weight; HBA, hydrogen bond acceptors; HBD, hydrogen bond donors.

**Table 3 molecules-28-00149-t003:** Pharmacokinetics and drug-like characteristics of carbohydrazide derivatives (**1a**–**5d**).

Compound Codes	Pharmacokinetics									Drug-Likeness			
	GI abs.	BBB pen.	P-gp sub.	CYP1A2	CYP2C19	CYP2C9	CYP2D6	CYP3A4	Log *K*_p_ (Skin Permeation, cm/s)	Ghose	Egan	Muegge	Bioavailability Score
				Inhibitors									
NL	H	Y	Y	N	N	N	Y	N	–7.43	Y	Y	Y	0.55
**1a**	H	N	Y	N	N	N	N	N	–6.87	0	0	0	0.55
**1b**	H	N	Y	N	N	N	N	N	–7.22	0	0	0	0.55
**1c**	H	N	Y	N	N	Y	N	N	–7.07	0	0	0	0.55
**1d**	H	N	Y	N	N	Y	N	N	–7.07	0	0	0	0.55
**2a**	H	N	Y	N	N	Y	N	N	–6.63	0	0	0	0.55
**2b**	H	N	Y	N	N	N	N	N	–6.99	0	0	0	0.55
**2c**	H	N	Y	N	N	Y	N	Y	–6.84	0	0	0	0.55
**2d**	H	N	Y	N	N	Y	N	Y	–6.84	0	0	0	0.55
**3a**	H	N	Y	Y	Y	Y	N	Y	–6.4	0	0	0	0.55
**3b**	H	N	Y	Y	N	N	N	Y	–6.75	0	0	0	0.55
**3c**	H	N	Y	Y	Y	Y	N	Y	–6.6	0	0	0	0.55
**3d**	H	N	Y	Y	Y	Y	N	Y	–6.6	0	0	0	0.55
**4a**	H	N	Y	N	N	N	N	N	–6.91	0	0	0	0.55
**4b**	H	N	Y	N	N	N	N	N	–7.26	0	0	0	0.55
**4c**	H	N	Y	N	N	Y	N	Y	–7.11	0	0	0	0.55
**4d**	H	N	Y	N	N	Y	N	Y	–7.11	0	0	0	0.55
**5a**	H	N	Y	N	N	N	N	N	-6.95	0	0	0	0.55
**5b**	H	N	Y	N	N	N	N	N	–7.29	0	0	0	0.55
**5c**	H	N	Y	N	N	Y	N	Y	–7.15	0	0	0	0.55
**5d**	H	N	Y	N	N	Y	N	Y	–7.15	0	0	0	0.55

Where: NL, Native ligand; GI abs., gastrointestinal absorption; BBB pen., blood-brain barrier penetration; P-gp sub., p-glycoprotein substrate; H, High; Y, Yes; N, No.

**Table 4 molecules-28-00149-t004:** Toxicity prediction of carbohydrazide derivatives (**1a**–**5d**).

Compound Codes	Parameters							
	LD_50_ (mg/kg)	Toxicity Class	Prediction Accuracy (%)	Hepatotoxicity (Probability)	Carcinogenicity (Probability)	Immunotoxicity (Probability)	Mutagenicity (Probability)	Cytotoxicity (Probability)
NL	800	4	23	I (0.60)	A (0.50)	A (0.80)	I (0.65)	I (0.71)
**1a**	711	4	54.26	A (0.64)	A (0.71)	I (0.98)	I (0.60)	I (0.72)
**1b**	1644	4	54.26	A (0.64)	A (0.66)	I (0.65)	I (0.62)	I (0.78)
**1c**	1880	4	54.26	A (0.61)	A (0.60)	A (0.55)	I (0.59)	I (0.80)
**1d**	1880	4	54.26	A (0.62)	A (0.60)	I (0.89)	I (0.59)	I (0.77)
**2a**	1000	4	54.26	A (0.59)	A (0.52)	I (0.96)	I (0.69)	I (0.76)
**2b**	1000	4	23	A (0.62)	A (0.52)	A (0.69)	I (0.70)	I (0.79)
**2c**	1000	4	23	A (0.61)	I (0.53)	A (0.84)	I (0.68)	I (0.77)
**2d**	1190	4	100	A (0.69)	I (0.62)	A (0.96)	I (0.97)	I (0.93)
**3a**	1000	4	54.26	A (0.59)	A (0.52)	I (0.93)	I (0.69)	I (0.76)
**3b**	1190	4	100	A (0.69)	I (0.62)	A (0.96)	I (0.97)	I (0.93)
**3c**	1000	4	23	A (0.61)	I (0.53)	A (0.91)	I (0.68)	I (0.77)
**3d**	1000	4	23	A (0.61)	I (0.53)	A (0.50)	I (0.68)	I (0.77)
**4a**	1000	4	23	A (0.60)	A (0.57)	I (0.97)	I (0.68)	I (0.77)
**4b**	1644	4	23	A (0.64)	A (0.55)	A (0.59)	I (0.67)	I (0.79)
**4c**	711	4	23	A (0.63)	I (0.51)	A (0.78)	I (0.66)	I (0.78)
**4d**	1880	4	23	A (0.63)	I (0.51)	I (0.76)	I (0.66)	I (0.78)
**5a**	1000	4	23	A (0.60)	A (0.57)	I (0.60)	I (0.68)	I (0.77)
**5b**	1644	4	23	A (0.64)	A (0.55)	A (0.97)	I (0.67)	I (0.79)
**5c**	711	4	23	A (0.63)	I (0.51)	A (0.98)	I (0.66)	I (0.78)
**5d**	1880	4	23	A (0.63)	I (0.51)	A (0.91)	I (0.66)	I (0.78)

Where: NL, Native ligand; I, Inactive; A, Active.

**Table 5 molecules-28-00149-t005:** The binding interactions of the most potent molecules with the target enzyme.

Active Amino Acid Residues	Atom from Ligand	Bond Length (Å)	Bond Type	Bond Category	Ligand Energy	Docking Scores
					(kcal/mol)	
Native Ligand						
TYR662	N-H	1.66907	Hydrogen Bond	Conventional Hydrogen Bond	447.3	−9.1
ARG125	Pi-Orbitals	4.39768	Electrostatic	Pi-Cation		
ARG358		3.52293				
ARG358		5.41244	Hydrophobic	Pi-Alkyl		
PHE357	Aromatic Carbon	3.79334				
**1b**						
GLU378	N	4.2858	Electrostatic	Attractive Charge	299.89	−9.4
GLU378	N	4.09602				
SER349	H	2.37918	Hydrogen Bond	Conventional Hydrogen Bond		
SER376	H	2.55398				
SER376	NH	2.862				
THR351	O	1.98084				
THR351	O	1.67832				
GLU378	C	3.06114		Carbon Hydrogen Bond		
THR350	Pi-Orbitals	3.1604		Pi-Donor Hydrogen Bond		
SER376		3.06845				
PHE396		4.9139	Hydrophobic	Pi-Pi T-shaped		
CYS394	Alkyl	5.29122		Pi-Alkyl		
VAL354		5.28734				
**2b**						
ARG560	NH	2.90874	Hydrogen Bond	Conventional Hydrogen Bond	343.49	−9.3
ASN562	H	2.81998				
LYS512	Pi-Orbitals	3.51422	Hydrophobic	Pi-Sigma		
ILE529		3.43887				
ARG560		3.75323				
PHE559		5.21502		Pi-Pi T-shaped		
PRO475	Cl	5.41082		Alkyl		
LYS512	Cl	4.09612				
ALA564	Pi-Orbitals	5.11889		Pi-Alkyl		
PRO475		4.72305				
PHE559		5.12368				
**3b**						
GLU205	N	5.12407	Electrostatic	Attractive Charge	340.69	−9.3
GLU205	N	5.23339				
HIS740	H	2.57758	Hydrogen Bond	Conventional Hydrogen Bond		
PHE357	Pi-Orbitals	3.79816	Hydrophobic	Pi-Pi Stacked		
TYR662		4.0191				
TYR547		4.68336				
TYR666		4.89163		Pi-Pi T-shaped		
SER630; TYR631		4.99633		Amide-Pi Stacked		
LYS554	Cl	4.43461		Alkyl		
VAL656	Pi-Orbitals	5.41552		Pi-Alkyl		
TYR547	Cl	4.69875				
TYR547	Cl	4.67797				
**4a**						
PHE559	F	3.12032	Hydrogen Bond;Halogen	Carbon Hydrogen Bond;Halogen (Fluorine)	316.76	−9.5
VAL558	F	3.51875	Halogen	Halogen (Fluorine)		
LEU514	Pi-Orbitals	4.68451	Hydrophobic	Pi-Alkyl		
PRO510		5.49372				
LYS512		4.13088				
ILE529		4.76619				
**5b**						
ILE102	H	2.81295	Hydrogen Bond	Conventional Hydrogen Bond	316.62	−9.6
HIS100	H	1.85943				
ILE102	O	2.2207				
ASN74	F	3.14132	Halogen	Halogen (Fluorine)		
LYS71	Pi-Orbitals	4.97015	Electrostatic	Pi-Cation		
TYR105		5.06176	Hydrophobic	Pi-Pi T-shaped		
ILE76		5.43513		Pi-Alkyl		
ILE102		5.10056				

**Table 6 molecules-28-00149-t006:** The 2D- and 3D-binding postures of the most potent molecules.

3D-Docking Poses	2D-Docking Poses
Native Ligand	
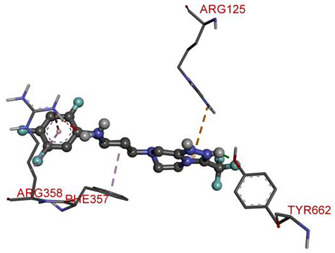	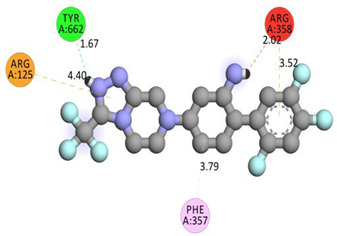
Compound **1b**	
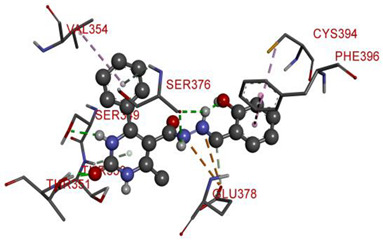	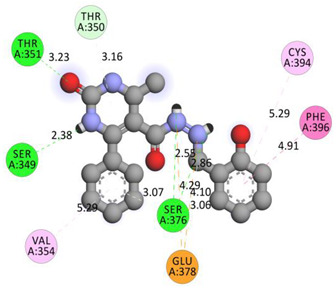
Compound **2b**	
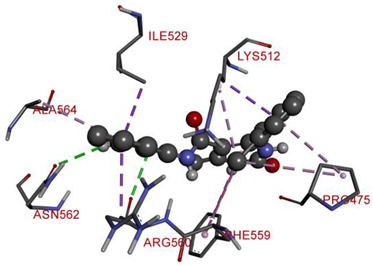	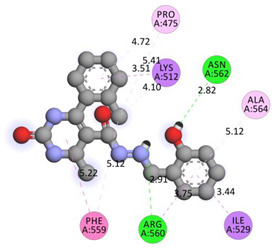
Compound **3b**	
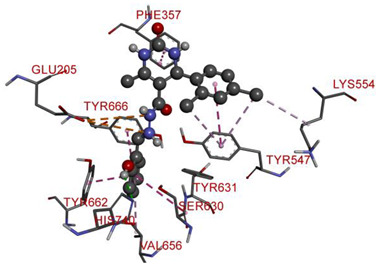	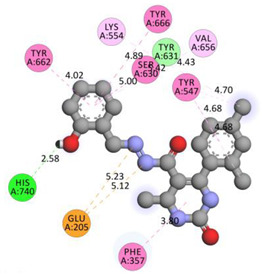
Compound **4a**	
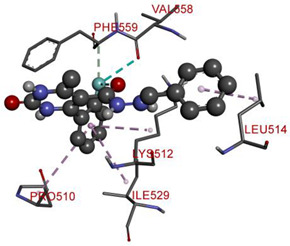	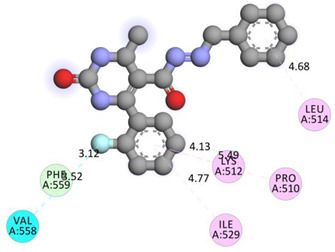
Compound **5b**	
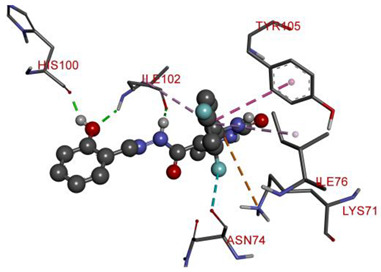	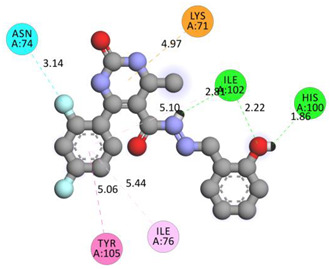

**Table 7 molecules-28-00149-t007:** In vitro enzyme assay of synthesized compounds.

Compound Code	% Inhibition at 50 µM	IC_50_ (µM)	Compound Code	% Inhibition at 50 µM	IC_50_ (µM)
**1a**	40.74 ± 0.38	58.95 ± 0.0012	**3c**	61.00 ± 0.08	33.73 ± 0.0024
**1b**	42.69 ± 0.28	54.73 ± 0.0025	**3d**	63.25 ± 0.05	32.38 ± 0.0021
**1c**	56.47 ± 0.37	39.72 ± 0.0027	**4a**	64.78 ± 0.33	30.47 ± 0.0025
**1d**	57.74 ± 0.38	38.26 ± 0.0050	**4b**	63.80 ± 0.16	31.75 ± 0.0027
**2a**	61.25 ± 0.16	32.41 ± 0.0033	**4c**	60.40 ± 0.08	34.94 ± 0.0016
**2b**	65.50 ± 0.08	31.93 ± 0.0029	**4d**	57.74 ± 0.38	38.62 ± 0.0025
**2c**	63.25 ± 0.05	33.97 ± 0.0025	**5a**	44.20 ± 0.28	65.31 ± 0.0021
**2d**	64.78 ±0.33	31.72 ± 0.0021	**5b**	70.70 ± 0.37	28.13 ± 0.0029
**3a**	61.25 ± 0.16	33.91 ± 0.0029	**5c**	58.54 ± 0.27	37.63 ± 0.0026
**3b**	61.00 ± 0.08	33.79 ± 0.0021	**5d**	62.28 ± 0.29	34.13 ± 0.0027
Sitagliptin	101.7 ± 0.09	0.018 ± 0.0012	---		

All values are expressed as the means of three independent determinations.

**Table 8 molecules-28-00149-t008:** Comparison of the effect of **1b**, **4c**, and **5b** on fasting blood glucose levels to that of the standard antidiabetic drug “sitagliptin”.

Treatment Group	Fasting Blood Glucose (mg/dL) ± SEM			
	On Day 1	On Day 7	On Day 14	On Day 21
Diabetic control	265.33 ± 6.27	269.66 ± 7.96	275.33 ± 1 0.04	280.00 ± 13.29
Sitagliptin (10 mg/kg)	240.16 ± 9.88	178.66 ± 6.92 ***	175.16 ± 8.61 ***	133.50 ± 11.80 ***
Compound **1b** (50 mg/kg)	261.00 ± 8.44	215.03 ± 3.29 *	187.05 ± 6.48 *	160.12 ± 6.18 *
Compound **5b** (50 mg/kg)	253.83 ± 4.49	182.00 ± 8.11 *	167.16 ± 6.23 **	157.33 ± 5.75 **
Compound **4c** (50 mg/kg)	264.66 ± 5.70	237.16 ± 9.01	194.83 ± 4.81 *	169.66 ± 8.53 *

The data are expressed as the mean ± S.E.M. (standard error of the mean) of six rats. * significantly differs from diabetic control group; the levels of significance are indicated as * *p* < 0.05, ** *p* < 0.01 and *** *p* < 0.001.

## Data Availability

Not applicable.

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
