# Peer review of "Virtual Screening, Synthesis, and Biological Evaluation of Some Carbohydrazide Derivatives as Potential DPP-IV Inhibitors"

_molecules, 2022, doi:10.3390/molecules28010149_

Round 1

Reviewer 1 Report

In the manuscript entitled "Virtual screening, synthesis, and biological evaluation of some 2 6-methyl-2-oxo-4-substituted-N'-[(E)substituted-methylidene]-13 ,2,3,4-tetrahydropyrimidine-5-carbohydrazide derivatives as 4 potential DPP-IV inhibitors" by Jadhav and col., the authors present an interdisciplinary study of new DPP-IV inhibitors as potential anti-diabetic drugs that they have designed, ranked according to their docking score, synthesized and evaluated in vitro, employing a selection of the best candidates in further animal studies. Overall, the results are promising, and the paper could be eventually published in Molecules. In my opinion, however, there is a big issue with the presentation of the information: the authors show four comprehensive tables referring to compounds 1a-5d WITHOUT showing the structures of such compounds first. In fact, their molecular structures have to be inferred from the 2D and 3D depictions of the interaction modes found by the docking calculations. Not even in the Materials and Methods sections do they show the chemical structures. I consider this issue must be corrected before the paper can be considered for publication. Because, otherwise, it is very confusing for the reader. Moreover, the authors fail to mention how they came up with these structures. Just a few lines in the introduction, commenting perhaps the similarities between their trial battery of compounds and the natural ligand should be enough. Also, I strongly suggest they replace the term "electron-repulsing" by the commonly accepted "electron-withdrawing". Finally, I would like the authors to comment on the fact that the best of their compounds (5b) exhibits an IC50 value that is >1400 times higher than the IC50 of Sitagliptin. Yet, in the animal studies (where they employ their new inhibitors at 5x the concentration of the natural ligand), the obtained profiles are pretty similar. How can the difference observed in the in vitro tests virtually disappear in the animal studies? Is it possible that the new batch of compounds are targeting some other enzymes also involved in the control of blood glucose levels?

Author Response

The detailed response to every query has been provided in the attached file

Reviewer 2 Report

In this manuscript, the authors report the virtual screening, synthesis, and biological evaluation of carbohydrazide derivatives that are potential DPP-IV inhibitors. The work is rather standard but well carried out. I think the paper is overly long and parts of the tables/figures could be moved to the SI. Perhaps a word of caution to the reader about the selection rules adopted by the authors could be added, for instance about the rule of five (https://www.nature.com/articles/nrd.2018.197).

Author Response

Detailed response to every query of Reviewer 2 has been provided in the attached file 

Reviewer 3 Report

The work on the “ Virtual screening, synthesis, and biological evaluation of some 6-methyl-2-oxo-4-substituted-N'-[(E)substituted-methylidene]-13 ,2,3,4-tetrahydropyrimidine-5-carbohydrazide derivatives as potential DPP-IV inhibitors” is a valuable perfect, and suitable contribution to be published in Molecules Journal after justifying some points.

·       The title showed be changed, no need to write the full IUPAC name of the main structure you can change it as “ Virtual screening, synthesis, and biological evaluation carbohydrazide derivatives as potential DPP-IV inhibitors” to be easier to read

·       The Affiliation Number 10 was not cited for any author??

1-     Abstract

·       you can improve the abstract and no need to write the full IUPAC name of the main structure so you can write carbohydrazide derivatives line 46-47

·       You can add a sentence regarding the chemical characterization methods of these compounds “ NMR, HRMS, IR “ to the abstract.

·       you can add the name of the diabetic control which was used

·       in the keywords I prefer to use the main family name carbohydrazide rather than write tetrahydropyrimidine.

2-     Introduction

·       The introduction is well written but may you can improve it by adding the problem statement (DM disease) in the first paragraph

·       You can add some statistical data regarding DM to the introduction from recent publications, https://doi.org/10.1038/s41598-022-07188-2, https://doi.org/10.1186/s13065-021-00766-x, as well as I recommended you to write a small paragraph regarding similar structures that showed similar biological activities.

3-     Results and discussion

·       you have to write a section regarding chemistry results, yields, HRMS and NMR main findings.

·       in the captions of the tables no need to write the full IUPAC name

·       I prefer to write the standard values of each Lipinski rule of five and vibers rule in table 1 to make a comparison

·       In table 5 may you can show the results of the most active compounds and transfer the others to the supplementary file

·       Line 208 in the 4 series the highest score was for 4a but the most active compound in vitro was 4c can you explain that??

·       in Table 6 where is the SD of the IC50 values???

·       can you explain in the main text why you used a dose for positive control different than your compounds

4-     Materials and Methods

·       add a small paragraph regarding the used chemicals, reagents, and instruments

·       regarding the Molinspiration calculation, Lipinski’s rule of five you add more recent references 

·     I could not find the NMR HRMS or IR spectrum can you provide it as a supplementary file ??

·       The Conclusion can be improved  

·       Control again all references as the journal style  

 Best wishes

Author Response

The detailed response to every query of Reviewer 3 has been provided in the attached file 

Round 2

Reviewer 1 Report

The authors have improved considerably their manuscript by incorporating the suggestions of all reviewers. However, the resulting new version looks rushed and more effort should be put into it before it is ready for publication. Particulary, the new bits of text that are marked in yellow require extensive English language revision, as some parts are difficult to understand. More importantly, regarding the characterization of the compounds, even if the authors claim they do not have access to the NMR spectra anymore, they should carefully rewrite these data according to this journal's policy, since there is no coherence in the order of the signals listed, or even the number of decimals for the chemical shift of each signal. This must be corrected.

For all these reasons, I cannot accept this manuscript for publication in its present form.

Author Response

Reviewer-01 comment:

The authors have improved considerably their manuscript by incorporating the suggestions of all reviewers. However, the resulting new version looks rushed and more effort should be put into it before it is ready for publication. Particularly, the new bits of text that are marked in yellow require extensive English language revision, as some parts are difficult to understand. More importantly, regarding the characterization of the compounds, even if the authors claim they do not have access to the NMR spectra anymore, they should carefully rewrite these data according to this journal's policy, since there is no coherence in the order of the signals listed, or even the number of decimals for the chemical shift of each signal. This must be corrected. For all these reasons, I cannot accept this manuscript for publication in its present form.

Response to Reviewer-1 comments:

We would like to thank the reviewer for his close observation and suggestions for the improvement of our manuscript. As suggested by the reviewer, English language editing has been done by a native English-speaking professional. Besides, we have also corrected the spectral data. In addition, the whole manuscript has been checked thoroughly and revised. All the changes in the manuscript are highlighted in yellow coloured.

We hope that you would be happy with the present form of the manuscript for publication. 
